# Clinician-created educational video for shared decision-making in the outpatient management of acne

Chih-Tsung Hung[1,2,3], Yi-Hsien Chen[1,3], Tzu-Ling Hung[1], Chien-Ping Chiang[1], Chih-Yu Chen[1,2], Wei-Ming Wang[1,2,4]*

**1** Department of Dermatology, Tri-Service General Hospital, National Defense Medical Center, Taipei, Taiwan, **2** Graduate Institute of Medical Sciences, National Defense Medical Center, Taipei, Taiwan, **3** Department of General Medicine, Tri-Service General Hospital, Taipei, Taiwan, **4** Vice Chairman, School of Medicine, National Defense Medical Center, Taipei, Taiwan

* ades0431@ms38.hinet.net

## Abstract

Shared decision-making (SDM) provides patient-centered care. However, the limited consultation time was the main factor hindering the application. Patient education is crucial in the process of SDM. The use of visual aids as health education materials is an effective way to improve patients' health literacy and medication adherence. This study aimed to determine the effectiveness of the clinician-created educational video of acne, accessed by patients during the waiting time, including knowledge level and satisfaction. This study was conducted in dermatology outpatient clinics and collected patient responses through electronic devices. During the waiting time, patients with acne would read educational pamphlets and complete the test first. Then, a clinician-created 8-minute educational video, as a patient decision aid (PDA), was accessed by patients using their own mobile smart devices, followed by a test and questionnaire about the satisfaction of the pamphlet and video. We enrolled 50 patients with acne, including 33 males and 17 females. The mean age is 25.55 ± 6.27 years old, ranging from 15 to 47 years old. About the patients' knowledge, the test score improved significantly after watching the video ($P < .001$). The same findings were observed in the subgroup analysis of gender and different age groups. A higher proportion of patients preferred the educational video over the pamphlet in both genders and different age groups. All patients agreed with the video helped them to understand the educational information and impressed them more than reading pamphlets. The application of clinician-created educational videos in patient education seems to be an efficient solution to implement SDM in the daily clinical work. Besides, we could remind patients to watch the video anytime when they were not sure about the treatment choices, side effects, or the precautions of medications.

## Introduction

Acne is a common skin disease and affects patients' life of quality a lot [1]. Oral isotretinoin, an isomer of retinoic acid, has been used to treat severe nodular acne, treatment-resistant moderate acne, and recalcitrant papulopustular rosacea [2, 3]. Isotretinoin has well-known

**Data Availability Statement:** We uploaded study's minimal underlying data as supplement file.

**Funding:** This study was supported by grants from Tri-Service Hospital Research Foundation (TSGH-E110219 and TSGH-E111201). The funders had no

role in study design, data collection, analysis, decision to publish, or preparation of the manuscript.

**Competing interests:** The authors have declared that no competing interests exist.

teratogenic effects, and the American Food and Drug Administration (FDA) mandated that all patients receiving isotretinoin are required to enroll in the iPLEDGE risk management program. However, there were 4,647 pregnancies reported to the FDA for patients taking isotretinoin after the introduction of iPLEDGE in 2006 [4]. In Taiwan, the Ministry of Health and Welfare requires all patients taking isotretinoin have to read and sign the informed consent after the discussion with dermatologists at clinics. Nonetheless, it is difficult to determine the patients' awareness level.

Shared decision-making (SDM) is a bidirectional process between healthcare professionals and patients to make treatment plans based on the medical evidence and patients' values [5]. SDM has shown the benefit in the improvement of patients' knowledge, satisfaction, and treatment adherence [6]. However, patients' level of health literacy affects their engagement in SDM a lot [7].

Health literacy is the ability to read, understand health information and make decisions regarding health in clinical practice [8]. Low literacy is related to poor health outcomes, lower treatment adherence [9], and frequent hospitalizations [10]. Several studies revealed the use of visual aids as health education materials, including pictograms and videos, is an effective way to improve patients' health literacy and medication adherence [11].

This study aimed to evaluate the use of a clinician-created educational video as a patient decision aid (PDA) to improve patients' knowledge about acne pathogenesis, treatment choices, isotretinoin mechanism, and side effects. We aimed to identify the patients' knowledge level, preference, and satisfaction with educational pamphlet and video.

## Materials and methods

### Study population and recruitment

The selection of the study participants is depicted in Fig 1. A single-center study was conducted involving 50 patients with moderate to severe acne. Patients were recruited from the authors' daily clinical practice, all of whom are licensed dermatologists working at an academic institution. Relatives or caregivers were recruited via the participating patients whose ages were under 20 years old. Inclusion criteria for all participants can access YouTube and watch the online video.

### Study design

The flowchart of the study design is described in Fig 1. According to our Health Insurance, the patients preparing to take isotretinoin must read and sign the standard inform consent (S1 File). After reading the education pamphlet (S2 File) and consent, they will complete the assessment (S3 File). In this study, we want to know the patients' awareness of acne and isotretinoin, and we designed an assessment of 5 quizzes reviewed by five dermatologists. Then, patients watched an 8-minute educational video on their mobile phones. After the video, they will complete the assessment and feedback questionnaire (S4 File). The questionnaire was designed as a 5-point Likert scale to evaluate the patients' perspective of educational video. One open-ended question was provided to patients and their family to express their opinions and feedback on the educational video. During the consultation, doctors can discuss with patients to make their treatment plans. All doctors involved in SDM were instructed before the implementation.

### Educational video

An 8-minute educational video was developed by Dr. Chih-Tsung Hung from the National Defense Medical Center in Taiwan. The video was edited followed the three steps as proposed by Ziade N. et al. in 2021 [12]. The video (which can be accessed via this link: Traditional Chinese

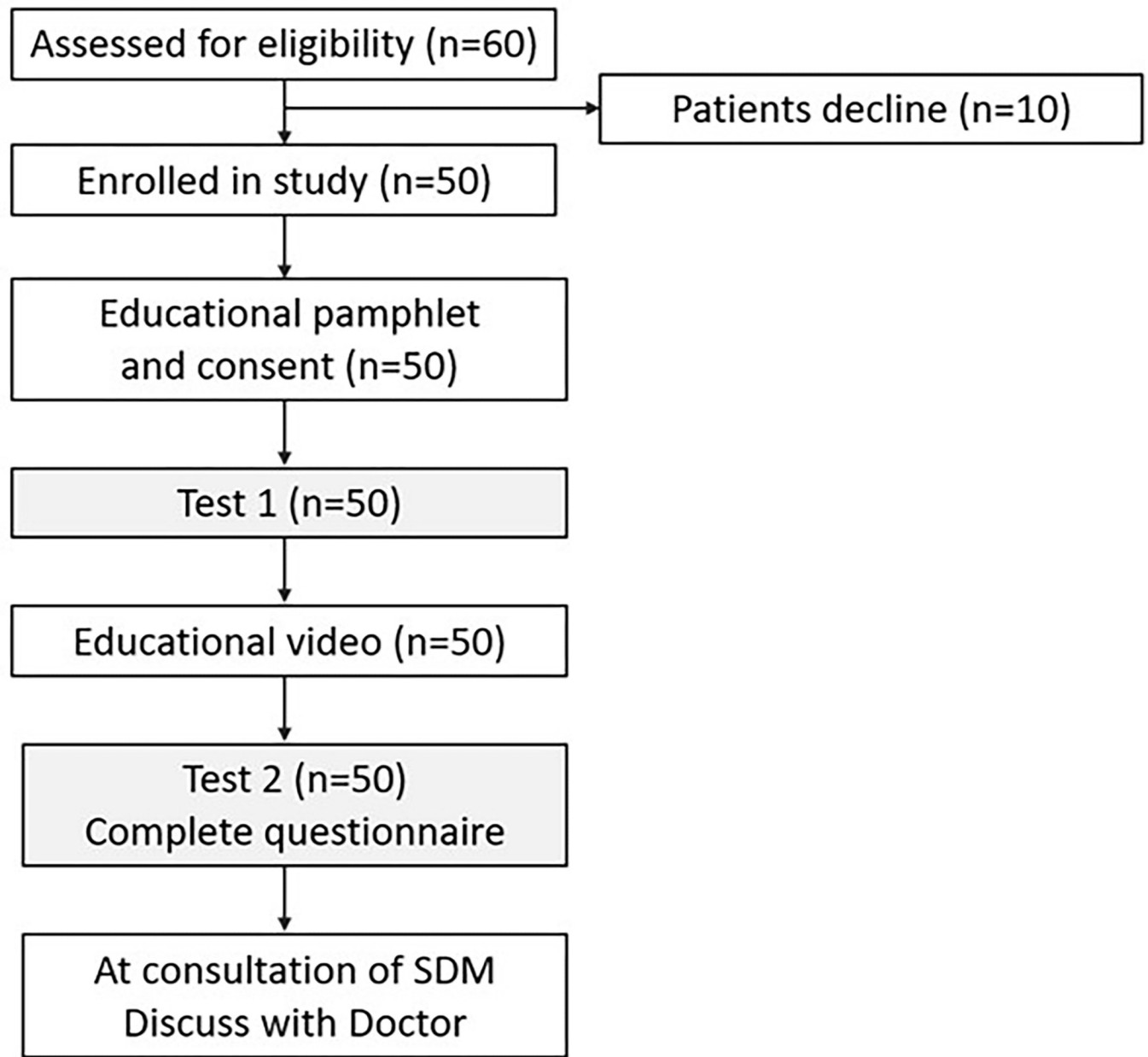

**Fig 1. The flowchart of study design and participants selection.** SDM, Shared decision-making.

version https://www.youtube.com/watch?v=L-cdfZHosCE; English version https://www.youtube.com/watch?v=5uTwAkPjAP4) is designed to improve patients' awareness of acne, treatment, and drug side effects. The topics addressed in the video and the spoken text were composed of acne pathogenesis, treatment choices, isotretinoin mechanism, and side effects. The video was reviewed and adapted by five dermatologists from the National Defense Medical Center in Taiwan.

## Statistical analysis

We used SPSS 22.0 software for Windows for all statistical analyses. To demonstrate the improvement of the medication awareness between educational pamphlets and the

educational video, a comparison of assessment scores was done by using a paired t-test. A *P* value less than 0.05 was considered statistically significant.

## Ethics

This study was reviewed and approved by the Institutional Review Board of Tri-Service General Hospital (TSGHIRB No.: B202105191). The study was conducted along with the code of Helsinki.

# Results

## Characteristics of patients

This study was conducted on patients with acne at the dermatology outpatient department. A total of 50 patients were enrolled, including 33 (66%) males and 17 (34%) females (Table 1). The mean age was 25.55 ± 6.27 years for the patients and the majority of the age group is between 20 to 25 years old, including 23 patients (14 males; 9 females). All patients read the educational pamphlets, completed the test, then watched the video, filled in the test and questionnaires. During the consultation, patients could discuss with the dermatologists if anything is remaining unclear (Fig 1).

## Test results after reading educational pamphlet and video

The test scores were collated and expressed as a mean score ± SD to find out the patients' learning efficiency (Table 2). The score improved significantly after watching the video (the score after reading the educational pamphlet was 81 ± 19.55, and the score after watching the video was 99 ± 4.79, *P* < .001). The same findings were observed in the subgroup analysis of gender and different age groups.

## Patients' satisfaction with educational pamphlet and video

Feedback was taken from 50 patients, and it is summarized (Fig 2). Among them, 41 (82%) patients preferred the video to understand acne mechanism and side effects of oral Isotretinoin, 7 (14%) patients preferred the pamphlet, and 2 (4%) patients felt the same. Subgroup

**Table 1. Characteristics of study patients in the baseline.**

| Variables | Total | |
|---|---|---|
| | **n** | **%** |
| **Total** | 50 | 100% |
| **Gender** | | |
| Male | 33 | 66% |
| Female | 17 | 34% |
| **Age (years)** | 25.55 ± 6.27 | |
| **Age groups (years)** | | |
| 15–19 | 7 (M/F = 7/0) | 14% |
| 20–25 | 23 (M/F = 14/9) | 46% |
| 26-29- | 11 (M/F = 6/5) | 22% |
| ≧30 | 9 (M/F = 7/2) | 18% |
| **Disease** | | |
| Acne | 50 | 100% |

M: male; F: female.

**Table 2. Test scores post-pamphlet and educational video.**

|  | Education pamphlet (n = 50) | Educational video (n = 50) | P value |
|---|---|---|---|
| **Score** | 81 ± 19.55 | 99 ± 4.79 | <0.001 |
| **Gender** |  |  |  |
| Male (n = 33) | 79 ± 19.96 | 99 ± 4.85 | <0.001 |
| Female (n = 17) | 86 ± 18.39 | 99 ± 5 | 0.007 |
| **Age groups score** |  |  |  |
| 15–19 yrs (n = 7) | 80 ± 16 | 100 | 0.018 |
| 20–24 yrs (n = 20) | 80 ± 17 | 97 ± 7 | <0.001 |
| 25–29 yrs (n = 14) | 89 ± 17 | 100 | 0.026 |
| $\geqq$30 yrs (n = 9) | 73 ± 28.28 | 100 | 0.022 |

Mean ± Standard Deviation; *P*: Paired t-test

analysis of gender and different age groups, the higher proportion of patients preferred the educational video.

## Patients' response to educational pamphlet and video

About the question (Q1): compared to the educational pamphlet, does the educational video impress you more about the acne mechanism, treatment, and side effects of oral isotretinoin? Feedback was taken from 50 patients, and it is summarized (Fig 3A). 38 (76%) patients strongly agreed and 12 (24%) patients agreed with it.

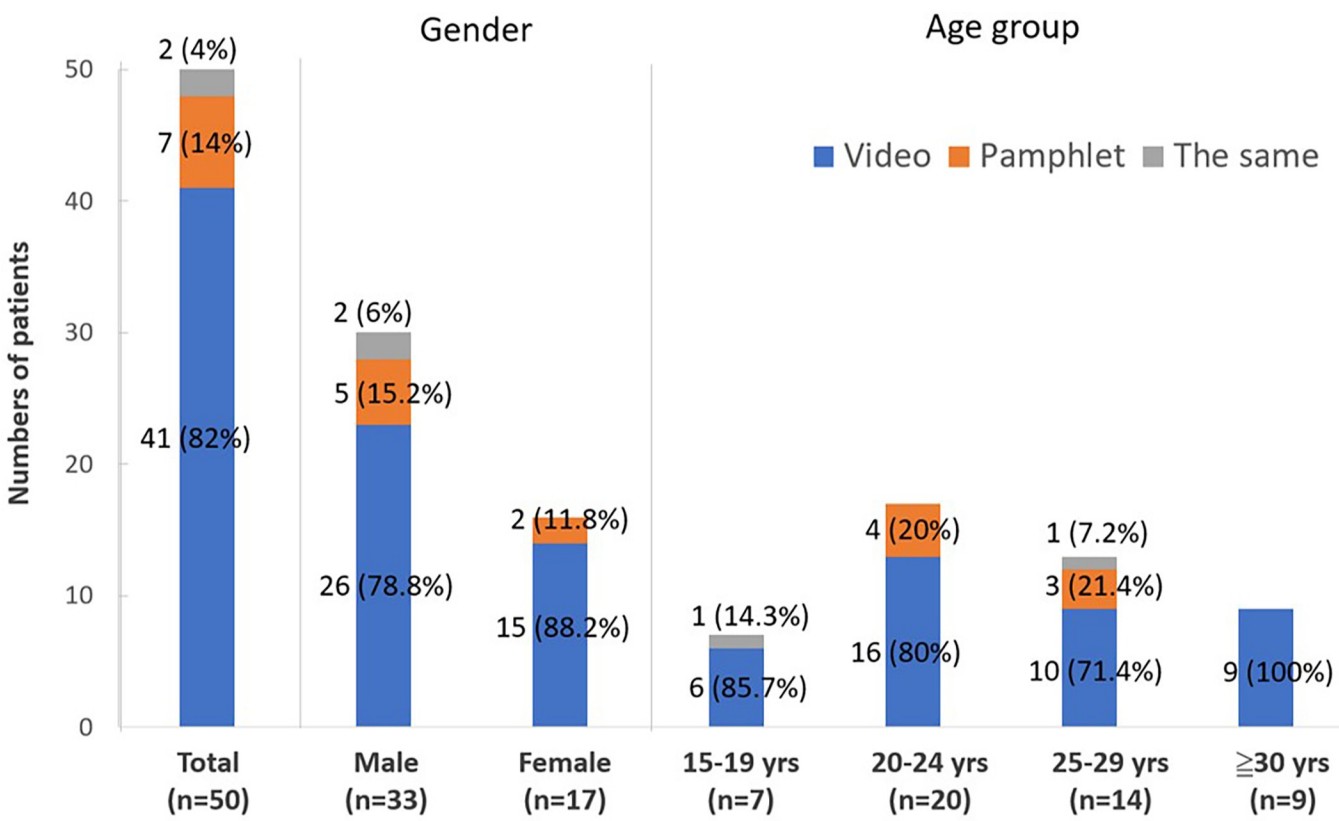

**Fig 2. Patients' satisfaction with educational pamphlet and video.**

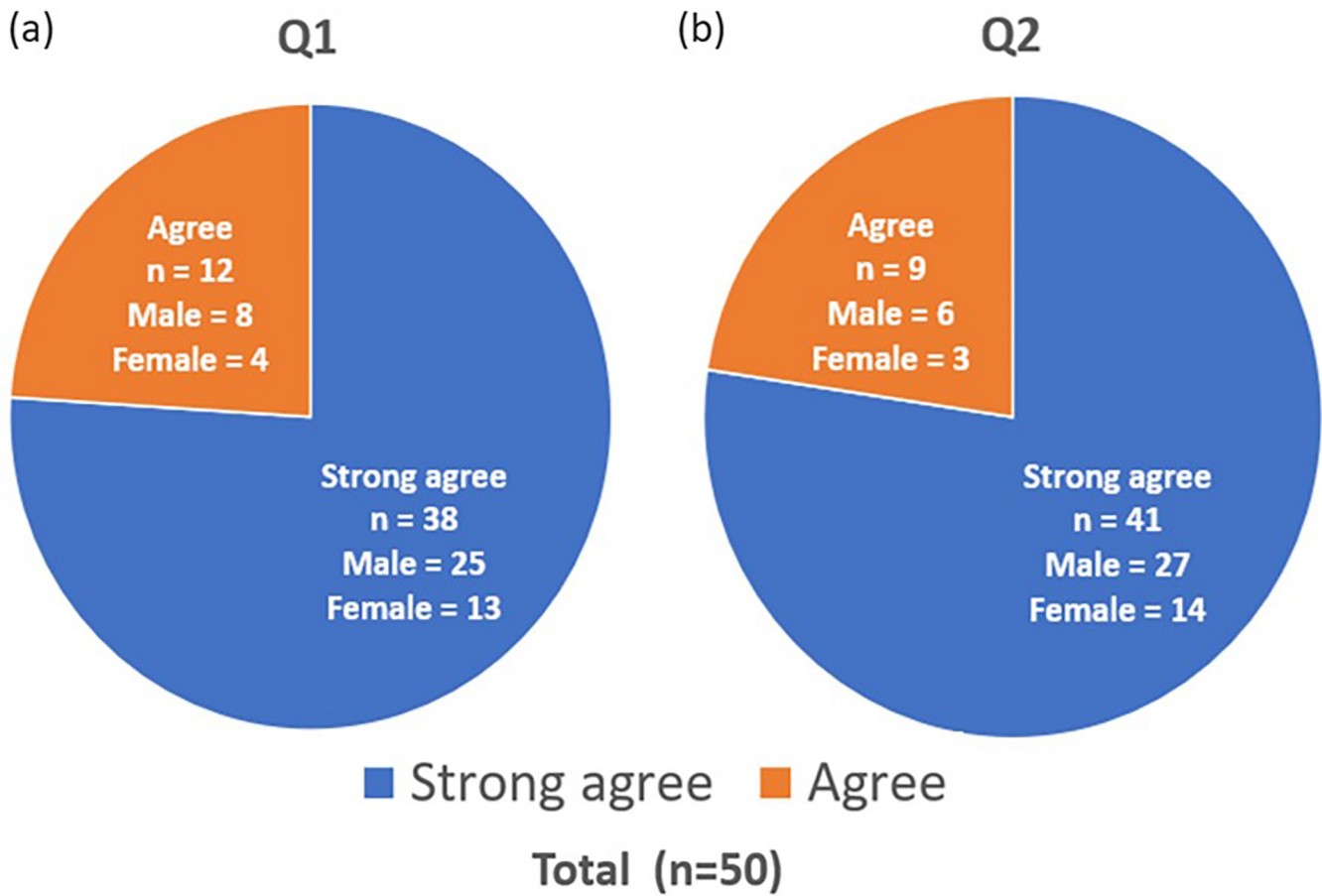

**Fig 3. Patients' response to educational pamphlet and video.** (a) Q1. Compared to the educational pamphlet, does the educational video impress you more about the acne mechanism, treatment, and side effects of oral isotretinoin? (b) Q2. Compared to the educational pamphlet, does the educational video help you to understand the acne mechanism, treatment, and side effects of oral isotretinoin more efficiently?

About the question (Q2): compared to the educational pamphlet, does the educational video help you to understand the acne mechanism, treatment, and side effects of oral isotretinoin more efficiently? Feedback was taken from 50 patients, and it is summarized (Fig 3B). 41 (82%) patients strongly agreed and 9 (18%) patients agreed with it.

There were 10 patients' qualitative responses to the educational video. 6 patients responded "very good", 3 patients responded "the content of video is clear and helped them understand efficiently", and one patient suggested that adding clinical picture of side effect about oral isotretinoin would impress him more.

## Discussion

In this study, we determined the efficacy of the educational video as a PDA supplement to the medical pamphlet to improve patients' awareness of acne and isotretinoin. The patients' knowledge level enhanced significantly after watching the video, and it made SDM perform efficiently during the consultation. This finding was observed in both males and females and the different age groups. Therefore, the educational video seems to be more effective at the enhancement of patients' awareness than the pamphlet irrespective of gender or age. The questionnaire is intended to be subjective to know the personal preferences of health education

that cannot be determined by the test score of patients. In our study, a higher proportion of patients favored the educational video than the pamphlet reading. Besides, compared to the pamphlet, most of them agreed that the video is more impressive and easier to understand the content.

SDM is composed of the following important elements: (1) it involves healthcare professionals and patients, (2) both sides share information, (3) both sides are in a course to build a consensus of the preferred treatment, and (4) a treatment agreement is reached and implemented [13]. Patient education plays an important role in the SDM [14]. It helps patients to be capable of asking more questions and expressing a greater desire for involvement during a consultation. In 2016, L. Tudor Car et al. used a priority-setting method to identify patient safety problems and solutions [15]. Inadequate patient education about their medication is one of the top three problems. The solution to the prevention is the improvement of patient education. In our study, we demonstrated the education video is a good supplement to the pamphlet on disease and medication. A higher score of awareness test was observed when the patients watched the video than that of reading pamphlet ($P < .001$). We proposed the educational video would be a solution to improve patient education.

SDM has been applied in dermatology for years [16], including melanoma [17], psoriasis [18, 19], and acne [19, 20], SDM provides patient-centered care, improves patient knowledge, satisfaction, and adherence to treatment [6]. However, there were several barriers to the application of SDM, including time, lack of training for healthcare professionals, lack of validated PDAs, and the inability of patients either due to low health literacy, lack of desire, or other factors [16]. The limited consultation time to implement SDM is the most mentioned barrier [21–23]. In this article, we used a clinician-created educational video as a PDA to let the healthcare professionals conduct SDM efficiently, and we reminded patients to watch the video anytime when they were not sure about the treatment choices, side effects, or the precautions of medications.

In agreement with previous studies, watching educational videos improved patient satisfaction and decreased patients' anxiety [24–26]. However, only a limited number of studies were designed to know the patients' knowledge improvement of medication safety after watching educational videos. In 2008, N Kinnane et al. demonstrated the use of video to standard chemotherapy education enhances patients' knowledge about the management of chemotherapy side effects [27]. All patients were satisfied with the video which helped them to remember the information given by the nurse. In 2021, JG Kovoor et al. showed clinician-created educational videos of atrial fibrillation in improving patient decision-making ability, long-term treatment adherence, and anxiety reduction [28]. In this study, we created the education video of acne as a PDA to implement SDM at clinics. Patents could access the video, quizzes, and the questionnaire during the waiting time by using Quick Response (QR) code. The results demonstrated not only the improvement of the patient knowledge but also high satisfaction with the video. Besides, healthcare professionals could discuss the treatment plans efficiently with patients during the consultation.

To the best of our knowledge, there is no study applying educational video as a PDA in dermatology to implement SDM. However, there were some limitations in our study. First, there is no validated questionnaire for acne knowledge. Therefore, the content is designed and reviewed by five different dermatologists. Besides, the questionnaire was also pretested by the patients and modified. Second, the main study population was Taiwanese. Thus, the ethnic discrepancy could exist. Third, we did not request the participants not to consult other resources, such as searching on the internet, during the filling in the assessment. However, the purpose of this study is to improve the patients' awareness of disease and medications. No matter which way the patients got the knowledge, it would be beneficial to increase patients' safety.

## Conclusion

This study demonstrated that educational video can be a PDA for patient education. The video helps healthcare professionals to perform SDM. It is an efficient way to share information with patients and their relatives, help them understand the benefits, harms, and possible outcomes of different options.

## Supporting information

**S1 File. Isotretinoin inform consent.**
(DOCX)

**S2 File. Education pamphlet of acne.**
(DOC)

**S3 File. Test quizzes of acne.**
(DOCX)

**S4 File. Questionnaire of educational video.**
(DOCX)

**S1 Data.**
(XLSX)

## Author Contributions

**Conceptualization:** Chih-Tsung Hung, Wei-Ming Wang.

**Data curation:** Chih-Tsung Hung, Tzu-Ling Hung, Chih-Yu Chen.

**Formal analysis:** Yi-Hsien Chen, Tzu-Ling Hung, Chien-Ping Chiang, Chih-Yu Chen, Wei-Ming Wang.

**Funding acquisition:** Chih-Tsung Hung.

**Investigation:** Yi-Hsien Chen, Chien-Ping Chiang.

**Methodology:** Chih-Tsung Hung, Yi-Hsien Chen, Tzu-Ling Hung, Chien-Ping Chiang, Chih-Yu Chen, Wei-Ming Wang.

**Project administration:** Chih-Tsung Hung.

**Resources:** Chien-Ping Chiang.

**Supervision:** Wei-Ming Wang.

**Validation:** Chien-Ping Chiang, Wei-Ming Wang.

**Writing – original draft:** Chih-Tsung Hung.

**Writing – review & editing:** Wei-Ming Wang.

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
