## [Decision Letter · Decision Letter 0]

10 Jun 2022

PONE-D-22-09574Clinician-created educational video for shared decision-making in the outpatient management of acnePLOS ONE

Dear Dr. Hung,

Thank you for submitting your manuscript to PLOS ONE. After careful consideration, we feel that it has merit but does not fully meet PLOS ONE’s publication criteria as it currently stands. Therefore, we invite you to submit a revised version of the manuscript that addresses the points raised during the review process.

We look forward to receiving your revised manuscript.

Kind regards,

Fabio Rinaldi

Academic Editor

PLOS ONE

Journal Requirements:

5. Please note that supplementary tables (should remain/ be uploaded) as separate "supporting information" files.

Additional Editor Comments:

The topic is interesting (patient education, the communication of the risks and benefits of a therapy) but the work does not give many useful indications except that communication via video is more appreciated than a pamphlet with the same explanations.

It may be interesting to evaluate the video shown in the link, but unfortunately the text is in Chinese

It would also be interesting to expand the discussion with some indications (if there are any)on the response of patients after watching the video, to know if their approach to therapy is different from those who do not undergo an educational video

I think it would be easier to accept the manuscript at least by evaluating the video in English and and adding the reaction of patients to the educational video and some indications for doctors who want to adopt this approach.

In general, I agree with observations from reviewers.

Please provide clarifications.

Reviewers' comments:

Reviewer's Responses to Questions

**Comments to the Author**

1. Is the manuscript technically sound, and do the data support the conclusions?

Reviewer #1: Partly

2. Has the statistical analysis been performed appropriately and rigorously? 

Reviewer #1: Yes

3. Have the authors made all data underlying the findings in their manuscript fully available?

Reviewer #1: No

4. Is the manuscript presented in an intelligible fashion and written in standard English?

Reviewer #1: Yes

5. Review Comments to the Author

Reviewer #1: -Video is not in english and so not easy to evaluated correctly -

-Authors must clarify if dermatologists instructed in the same way the patients. Were they instructed before? This coul be a bias of the study.

- Did authors make a power analysis fo choising the number of subjects to be tested?

- Video was editing following some standard guideline for patient education? If yes, please clarify in the text

- Main limitations of the sutdies must be highlighted.

6. PLOS authors have the option to publish the peer review history of their article (what does this mean?). If published, this will include your full peer review and any attached files.

Reviewer #1: No

---

## [Author Response · Author response to Decision Letter 0]

21 Jun 2022

Reviewer

Comments to the Author

Reviewer #1: -

1. Video is not in english and so not easy to evaluated correctly.

Response:

Thanks for your kind comment and reminder. We provided an educational video in English in the article (page 8, line 4). 

2. Authors must clarify if dermatologists instructed in the same way the patients. Were they instructed before? This could be a bias of the study.

Response:

Thanks for your kind comment and reminder. The content consisted of the treatment guide for acne in Taiwan. Besides, the video was reviewed by another 5 dermatologists, and the content was revised 3 times. All doctors involved in SDM were instructed before the implementation. 

3. Did authors make a power analysis fo choising the number of subjects to be tested?

Response:

Thanks for your kind comment and reminder. A two-tailed 99% confidence interval is ± 2.6 standard error (S.E.) wide. For a ±1 logit interval, this S.E. is ±1/2.6 logits, this gives a minimum sample in the range 4*(2.6)2 < N < 9*(2.6)2, i.e, 27 < N < 61, depending on targeting. Thus, a sample of 50 well-targeted examinees is conservative for obtaining useful.[1]

4. Video was editing following some standard guideline for patient education? If yes, please clarify in the text.

Response:

Thanks for your kind comment and reminder. The patient education video was edited followed the three steps as proposed by Ziade N. et al. in 2021 (page 8, line 1-2).[2] 

5. Main limitations of the sutdies must be highlighted.

Response:

Thanks for your kind comment and reminder. We revised the limitations in our article (page 15, line 13-19; page 16, line 1-4).

Reference

1. Khan MI. Recovery and stability of item parameter and model fit across varying sample sizes and test lengths in Rasch analysis with small sample. Social Science International. 2014;30(1):43.

2. Ziade N, Arayssi T, Elzorkany B, Daher A, Karam GA, Jbara MA, et al. Development of an Educational Video for Self-Assessment of Patients with RA: Steps, Challenges, and Responses. Mediterr J Rheumatol. 2021;32(1):66-73. Epub 20210304. doi: 10.31138/mjr.32.1.66. PubMed PMID: 34386703; PubMed Central PMCID: PMCPMC8314883.

---

## [Editor Report · Decision Letter 1]

24 Jun 2022

Clinician-created educational video for shared decision-making in the outpatient management of acne

PONE-D-22-09574R1

Dear Dr. Hung,

We’re pleased to inform you that your manuscript has been judged scientifically suitable for publication and will be formally accepted for publication once it meets all outstanding technical requirements.

Kind regards,

Fabio Rinaldi

Academic Editor

PLOS ONE

Additional Editor Comments (optional):

Ok for publications given the answers the Authors
---

## [Editor Report · Acceptance letter]

29 Jun 2022

PONE-D-22-09574R1 

Clinician-created educational video for shared decision-making in the outpatient management of acne 

Dear Dr. Hung:

I'm pleased to inform you that your manuscript has been deemed suitable for publication in PLOS ONE. Congratulations! Your manuscript is now with our production department. 

Kind regards, 

on behalf of

Dr. Fabio Rinaldi 

Academic Editor

PLOS ONE